# X-ray structure of a carpet-like antimicrobial defensin–phospholipid membrane disruption complex

Michael Järvå[1], Fung T. Lay[1], Thanh Kha Phan[1], Cassandra Humble[1], Ivan K.H. Poon[1], Mark R. Bleackley [1], Marilyn A. Anderson [1], Mark D. Hulett[1] & Marc Kvansakul [1]

Defensins are cationic antimicrobial peptides expressed throughout the plant and animal kingdoms as a first line of defense against pathogens. Membrane targeting and disruption is a crucial function of many defensins, however the precise mechanism remains unclear. Certain plant defensins form dimers that specifically bind the membrane phospholipids phosphatidic acid (PA) and phosphatidylinositol 4,5-bisphosphate, thereby triggering the assembly of defensin–lipid oligomers that permeabilize cell membranes. To understand this permeabilization mechanism, here we determine the crystal structure of the plant defensin NaD1 bound to PA. The structure reveals a 20-mer that adopts a concave sheet- or carpet-like topology where NaD1 dimers form one face and PA acyl chains form the other face of the sheet. Furthermore, we show that Arg39 is critical for PA binding, oligomerization and fungal cell killing. These findings identify a putative defensin–phospholipid membrane attack configuration that supports a longstanding proposed carpet mode of membrane disruption.

---

[1] Department of Biochemistry and Genetics, La Trobe Institute for Molecular Science, La Trobe University, Melbourne, Victoria 3086, Australia. Correspondence and requests for materials should be addressed to M.D.H. (email: M.Hulett@latrobe.edu.au) or to M.K. (email: M.Kvansakul@latrobe.edu.au)

Defensins are among the largest and best characterized family of cationic antimicrobial peptides (CAPs) that are produced by essentially all plant and animal species as an important component of their innate immune systems. These small (<60 amino acids), cationic, and cysteine-rich peptides can be expressed constitutively or inducibly, and exhibit broad antimicrobial activity. Furthermore, several plant and animal defensins show toxicity towards tumor cells[1–3]. Defensins have multiple mechanisms of action, including modulation of immune responses[4,5], triggering the production of reactive oxygen species[6,7], cell cycle interference[8], inhibition of enzymes[9–11] and ion channel activity[11–13], as well as the disruption of membranes[7,14,15]. Plant defensins can also play roles in root development and plant reproduction, underscoring the diversity of biological functions that can be mediated by this family of molecules[6,16].

A key function of defensins during innate defense is their ability to permeabilize and disrupt cell membranes. The mechanisms of action underlying membrane disruption by CAPs, including defensins, have not been fully defined and several putative modes of membrane attack have been proposed, including the (i) carpet, (ii) barrel-stave and (iii) toroidal-pore models[3,17]. Recently we demonstrated that the ability of certain plant defensins to kill fungal and tumor cells is dependent on the engagement of specific membrane phospholipids, which trigger the formation of large defensin–phospholipid oligomers[18–21]. Intriguingly, the phospholipid binding sites as well as the molecular architecture of the resultant oligomeric complexes that mediate membrane permeabilization are strikingly different. *Nicotiana alata* defensin 1 (NaD1) complexes with phosphatidylinositol 4,5-bisphosphate (PIP$_2$), forming a horseshoe-like arc comprising seven dimers and 14 PIP$_2$ molecules[18]. In contrast, a complex of the *Nicotiana suaveolens* defensin 7 (NsD7) with phosphatidic acid (PA) adopts a radically different topology comprising a coiled defensin–PA double helix[19]. These data demonstrate that different phospholipids trigger the formation of discrete defensin–phospholipid complexes with unique topologies. The interaction of these defensins, as well as other plant defensins including NaD2[22], TPP3[20], MtDef4[23], MtDef5[24], and HsAFP1[25], with PIP$_2$ or PA appears to be important for fungal cell killing. Furthermore, the interaction of Psd1[26], RsAFP2[27], and DmAMP1[28,29] with sphingolipids such as glycosylceramides and mannosyl diinositolphosphoryl ceramides has been linked with fungal cell killing, although whether these interactions also result in the formation of defensin–lipid complexes has not been determined. Taken together, these observations demonstrate that plant defensins recognize an array of different lipids for protection against fungi.

The ability to form oligomers is not limited to plant defensins, with oligomerization being an important mechanism in some human defensin innate immune functions. Indeed, human α-defensin 6 (HD6) and human β-defensin 1 (HBD-1) oligomerize to form nanonets that entangle bacteria and restrict their mobility[30,31]. However, the molecular cues or ligands that trigger net formation for HD6 and HBD-1 remain to be determined. Human β-defensin 3 (HBD-3) binds a range of phospholipids, including PIP$_2$, that is important for tumor cell permeabilization[15]. However, for each of these human defensins, whether the binding of phospholipids mediates oligomerization, has not been defined.

Despite the identification of phospholipids as key mediators and the crucial role of oligomer formation for target membrane attack, the molecular architecture of a membrane encounter complex of defensins and phospholipids has not been defined. To address this, we determined the crystal structure of a NaD1–PA complex to 2.5 Å. The structure reveals a near-flat, carpet-like oligomeric complex, which we termed the 'membrane disruption complex' (MDC), where a monolayer of NaD1 molecules engages PA molecules on a single side of the complex, thus mimicking the manner in which NaD1 would encounter PA molecules of target membranes. Structure-guided mutagenesis reveals Arg39 as a key residue for the cooperative PA-mediated oligomerization and fungal growth inhibition. These findings provide high-resolution structural evidence for the formation of a carpet-like configuration by a defensin during the initial stages of membrane encounter to engage target phospholipids and form oligomeric complexes that ultimately lead to membrane permeabilization.

## Results

**Crystal structure of NaD1–PA complex.** Since binding of PA has been previously shown to be important for membrane permeabilization and fungal cell killing by plant defensins[19,23,25], we examined the ability of NaD1 to bind PA and form oligomeric complexes using chemical crosslinking or negative stain TEM (Fig. 1a,b). In the presence of PA, NaD1 formed large oligomeric assemblies as shown by the distinct laddering after treatment with BS$^3$ as well as on non-reducing SDS-PAGE (Supplementary Figure 1). Importantly, native NaD1–PA complexes were visualized as long fibrils by TEM in the absence of any crosslinker (Fig. 1b). To understand the structural basis for PA-mediated oligomerization of NaD1 we crystallized an NaD1–PA complex and determined its structure to a resolution of 2.5 Å (Table 1). Clear and continuous electron density could be observed for all 20 NaD1 molecules, which allowed the identification of ten NaD1 dimers in the asymmetric unit that adopted a cationic grip configuration[18] harboring one or two PA molecules (Fig. 2a,b, Fig. 3a, Supplementary Figure 2 and 3). The ten dimers are arranged in a configuration that results in the formation of a mildly curved, carpet-like structure which we refer to as the membrane disruption complex (MDC), with two such 20-mers forming a large 40-meric ellipsoid complex with a total of 28 PA molecules (Fig. 3b,c). The MDC comprises three small arcs comprising three or four NaD1 dimers, respectively, in a 3–4–3-dimer configuration (Fig. 4a,b). The cationic grip of each NaD1 dimer, and the associated phospholipid tails of the bound PA molecules, are oriented towards the center of the ellipsoid,

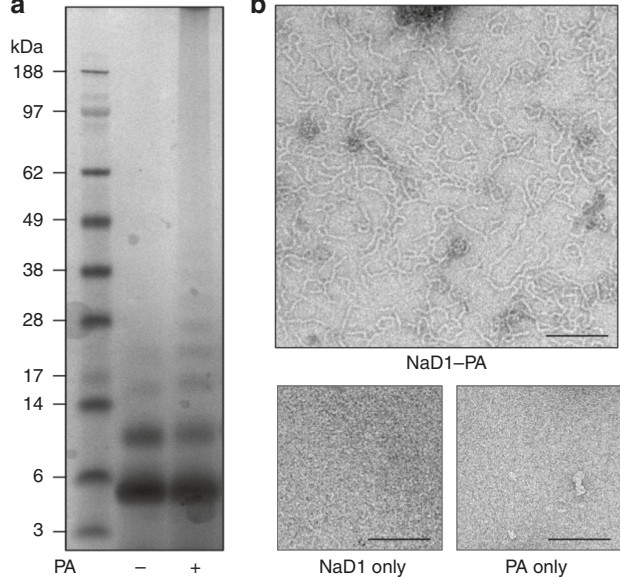

**Fig. 1** NaD1 forms oligomers with PA. **a**, **b** The ability of wild-type NaD1 to form multimeric complexes in the presence of PA as determined by (**a**) protein–protein crosslinking with BS$^3$ and visualized by SDS-PAGE, and by (**b**) transmission electron microscopy. Scale bars represent 200 nm

**Table 1 Data collection and refinement statistics**

|  | NaD1–PA |
| --- | --- |
| **Data collection** | |
| Space group | C 2 2 2₁ |
| No. of molecules in AU | 20 + 14 |
| **Cell dimensions** | |
| $a, b, c$ (Å) | 114.22 154.93 136.00 |
| $\alpha, \beta, \gamma$ (°) | 90.00, 90.00, 90.00 |
| Wavelength (Å) | 0.9537 |
| Resolution (Å)[a] | 47.06-2.50 (2.60-2.50) |
| $R_{sym}$ or $R_{merge}$[a] | 0.098 (0.612) |
| $I / \sigma I$[a] | 7.6 (1.5) |
| CC(1/2)[a] | 0.992 (0.630) |
| Completeness (%)[a] | 98.4 (99.8) |
| Redundancy[a] | 2.9 (2.9) |
| Wilson B-factor (Å²) | 32.8 |
| **Refinement** | |
| Resolution (Å) | 44.47-2.50 |
| No. of reflections | 41,062 |
| $R_{work} / R_{free}$ | 0.2031/0.2519 |
| **No. of non-hydrogen atoms** | |
| Protein | 7380 |
| Ligand/ion | 475 |
| Water | 451 |
| **B-factors** | |
| Protein | 42.42 |
| Ligand/ion | 64.20 |
| Water | 36.43 |
| **r.m.s. deviations** | |
| Bond lengths (Å) | 0.007 |
| Bond angle (°) | 1.10 |
| **Ramachandran plot (%)** | |
| Favored | 95.87 |
| Allowed | 4.13 |
| Disallowed | 0.00 |

[a]Values in parentheses are for highest-resolution shell

creating a hydrophobic core of encapsulated PA acyl chains within the center of the ellipsoid (Fig. 3b, Supplementary Movie 1). Since 20 NaD1 molecules are present in the MDC, a total of 20 PA binding sites were expected, however, in the structure only 14 PA molecules could be placed in the electron density (Fig. 4a,b).

**The PA binding site reveals a cooperative binding mode.** All 14 PA binding sites identified in the MDC are identical, and are formed by residues from the signature SKILRR motif in the β2–β3 loop of NaD1[32] (Fig. 4c,d). A single PA molecule makes contact with Ser35, Ile37, Leu38, and Arg40 from one NaD1 dimer, which enable Lys36 and Arg39 from a neighboring dimer to form hydrogen bonds with the phosphate head group via three water molecules that are conserved throughout the 14 PA binding sites (Supplementary Figure 4).

Within the MDC a total of seven dimer–dimer interfaces can be identified, which are further subdivided into two types of interfaces, designated type A and B. The central arc in the 3–4–3 assembly consists of two identical tetramers (rmsd = 0.204 Å), with each tetramer engaging two PA molecules each. Within each tetramer, the direct dimer–dimer interactions are primarily via cooperative binding of PA (discussed below), and a single hydrogen bond between Glu6 from one dimer and the backbone of Lys17 from an adjacent dimer (Supplementary Figure 5a). This interface is designated as type A. The two tetramers within the central arc are not stabilized by PA-mediated interactions, but instead associate via a more extensive hydrogen bond network consisting of Glu6–Lys17, and Arg40 to the backbones of Cys34 and Ser35 (Supplementary Figure 5b). This interface is designated as type B. The two smaller 3-dimer arcs that form the side of the MDC are near identical (rmsd = 0.228 Å). In contrast to the central arc where each dimer bound only one PA molecule, the 3-dimer arcs bind two PA molecules in two out of the three dimers (Fig. 4b), with the third dimer only featuring a single bound PA molecule. As with the tetrameric configuration in the central 4-dimer arc, the three dimers in the smaller peripheral arcs associate with each other via a type A interface via hydrogen

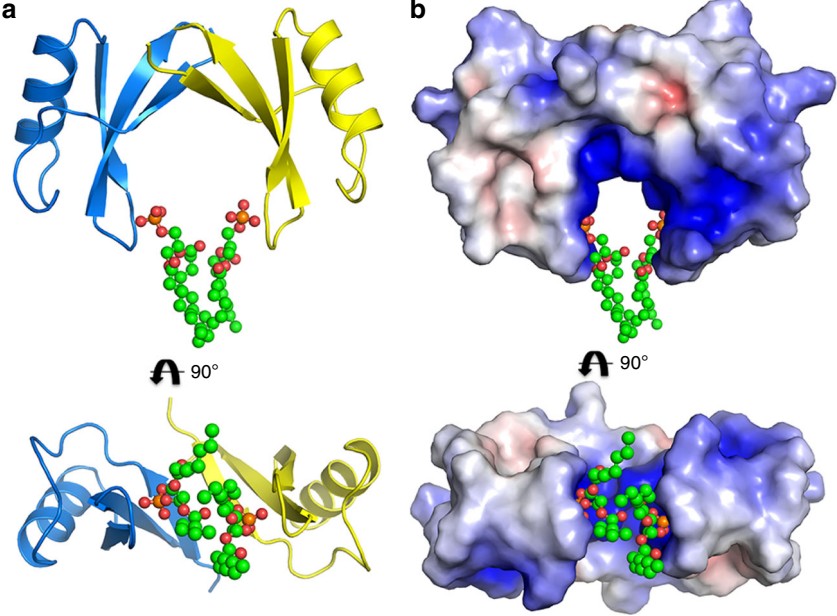

**Fig. 2** Crystal structure of the NaD1–PA dimer. **a** The NaD1 dimer in its cationic grip configuration (blue and yellow cartoon representation) bound to two molecules of PA (ball and stick with carbons in green, oxygen in red and phosphorus in orange). **b** Electrostatic surface representation of **a** highlighting the cationic charge in the grip cavity, which engage the head groups of bound phospholipids

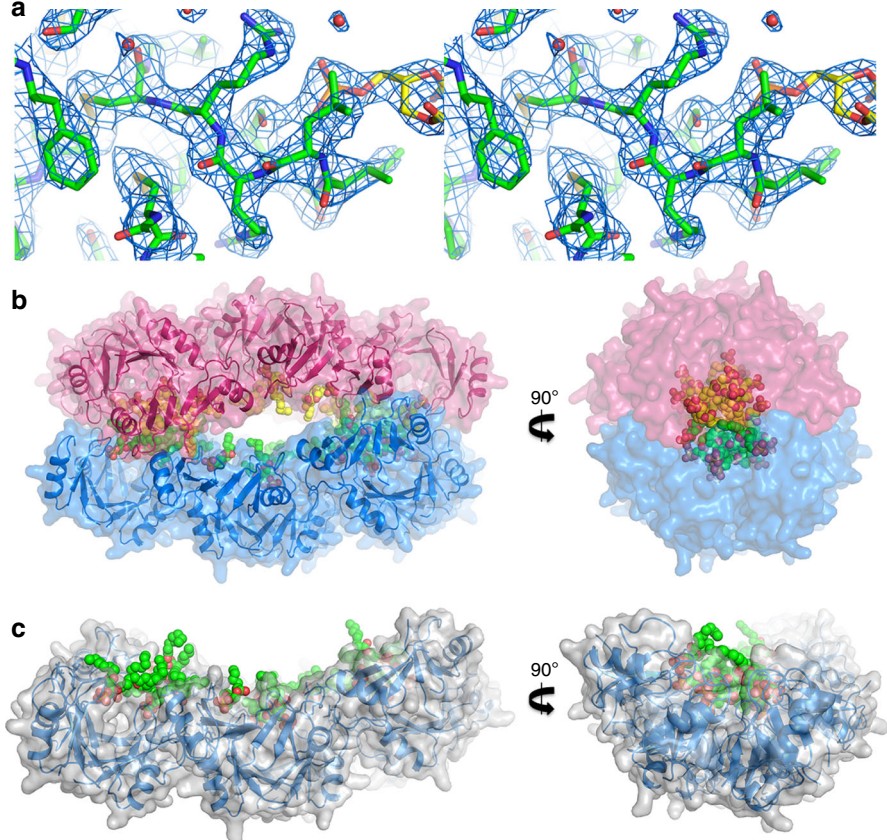

**Fig. 3** Crystal structure of the NaD1–PA complex. **a** Representative stereo view of the electron density maps for NaD1–PA structure. The 2Fo–Fc map is shown as a blue mesh contoured at 1.5σ. Protein carbon atoms are colored green while the PA carbon atoms are colored yellow. Oxygen is colored red, phosphorous orange, nitrogen blue, and sulfur yellow. **b** The 20-meric NaD1–PA complex (blue cartoon representation with gray transparent surface rendering) harboring 14 bound PA molecules (shown as spheres with carbon in green, oxygen in red, phosphorus in orange), as viewed from the front (left) and from the side (right). **c** Two 20-meric complexes in blue and pink cartoon/surface representation, with the PA molecules bound to the top or lower halves, indicated as green/red or yellow/red spheres, respectively. The front view (left) reveals the slight curvature of the complexes, while the side view (right) shows the hydrophobic core created from the acyl chains of bound PA facing inwards towards the center of the complex

bonds between Glu6 and Lys17 from adjacent dimers (Supplementary Figure 5a). Intriguingly, assembly of the MDC from its constituent 3-dimer and 4-dimer arcs is achieved almost exclusively via hydrophobic interactions where Ile13 and Ile15 from the β1-α1 loop of NaD1 form continuous hydrophobic interfaces along the sides of each arc (Fig. 5a). This configuration of Ile residues enables the formation of zipper-like structures to hold the MDC together (Fig. 5b).

**The role of Lys36 and Arg39**. To understand the importance of key interactions in the NaD1–PA complex, we compared PA-mediated oligomerization for wild-type with Lys36 and Arg39 point mutants of NaD1 using chemical crosslinking and TEM. In contrast to wild-type NaD1 and NaD1(K36E), the NaD1(R39A) mutant lost the ability to form oligomers with PA, while still maintaining PIP$_2$-mediated oligomerization (Fig. 6a,b, Supplementary Figure 6). To establish if loss of PA-specific oligomerization impacted the antimicrobial activity of NaD1 we then examined the ability of the three proteins to inhibit growth of a clinical isolate of the fungal pathogen *Candida albicans* (LTUMC001). Wild-type NaD1 displayed an IC$_{50}$ of $1.9 \pm 0.2$ μM, whereas NaD1(K36E) and NaD1(R39A) showed higher IC$_{50}$ at $2.5 \pm 0.1$ μM and $4.5 \pm 0.1$ μM, respectively (Fig. 7a), indicative of an attenuated ability to inhibit growth of *C. albicans*. We next examined whether the NaD1-induced growth inhibition of *C. albicans* is due to fungal killing by measuring uptake of the

membrane impermeable nucleic acid dye propidium iodide as well as colony forming assays, in the presence of NaD1. We examined *C. albicans* LTUMC001 and an additional two clinical isolates ATCC10231 and ATCC90028, which all displayed significant PI uptake when treated with low μM concentrations of wild-type NaD1 (Fig. 7b) and substantial reductions of colony forming units (Supplementary Figure 7). To determine if the NaD1(R39E) and NaD1(K36E) mutants had impaired ability to kill *C. albicans*, we then examined their effect on PI uptake in the three test isolates. Both mutants showed significantly reduced ability to permeabilize all three *C. albicans* isolates, with essentially no or low activity against ATCC10231 and ATC90028, although they retained some activity against LTUMC001, which was still significantly reduced when compared to wild-type NaD1 (Fig. 7c). Similar results were observed for the number of colony forming units (Supplementary Figure 7). We then confirmed the PI uptake results by performing live confocal laser scanning microscopy, where we observed rapid PI uptake in *C. albicans* (LTUMC001) when treated with wild-type NaD1, whereas both NaD1(K36E) and NaD1(R39A) showed substantially delayed and reduced PI uptake (Fig. 7d).

**Discussion**

An understanding of the mechanism by which CAPs such as defensins are able to cause membrane permeabilization and cell death is important for their development as antimicrobial and

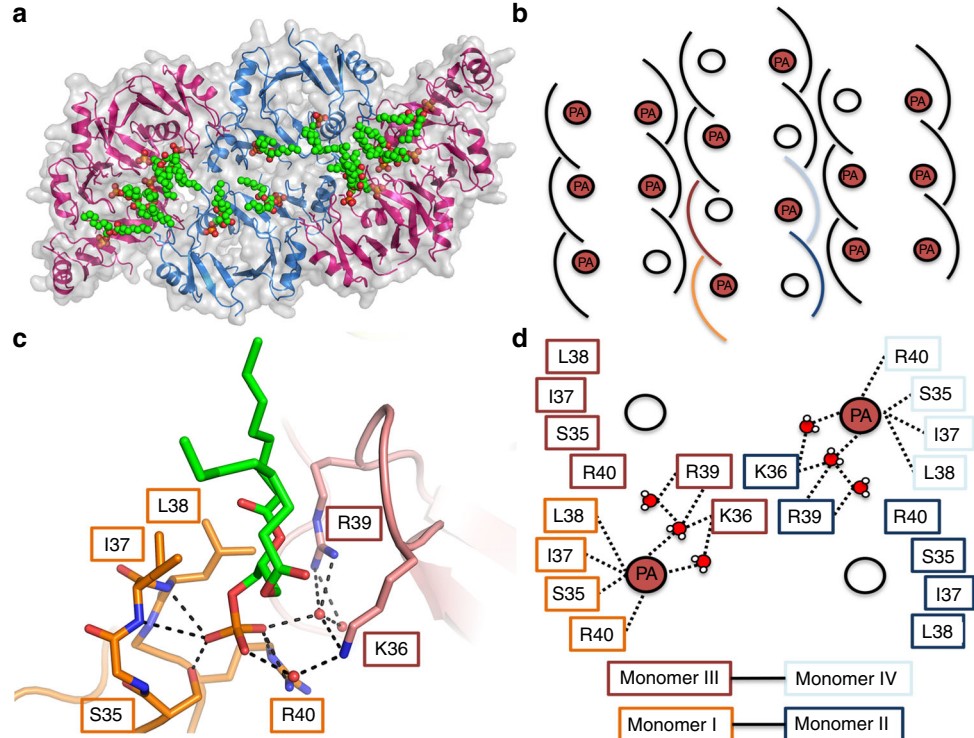

**Fig. 4** The NaD1–PA complex contains 14 PA molecules bound in identical NaD1-dimer binding sites. **a** The 20-meric complex (cartoon representation with gray transparent surface rendering, with the two 3-dimer arcs in pink, and the central 4-dimer arc in blue) harboring 14 bound PA molecules (shown as spheres with carbon in green, oxygen in red, phosphorus in orange), as viewed from the bottom. **b** A schematic of the three constituent arcs of the 20-meric complex. Each dimer engages one or two PA molecules (red circles), with the full complex binding 14 PA molecules in total. **c** A representative NaD1–PA binding site, with PA (shown as sticks) located within the cationic grip. One monomer of the dimer (orange) engages the PA phosphate head group through Ser35, Ile37, Leu38, and Arg40, while Arg39 and Lys36 (pink) of a single chain from a neighboring dimer are recruited through a hydrogen bond network involving three water molecules that are conserved in each binding site. **d** A schematic of how two dimers from **c** cooperatively bind two PA molecules

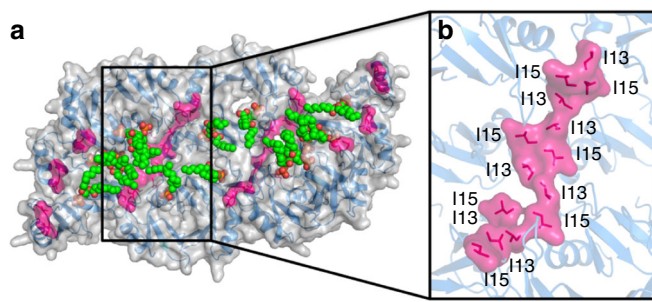

**Fig. 5** The arc–arc interactions in the membrane disruption complex are mediated via an isoleucine zipper. **a** The 20-meric complex (blue cartoon representation with gray transparent surface rendering) in the asymmetric unit with its 14 bound PA molecules (spheres with carbon in green, oxygen in red, phosphorus in orange). The residues Ile13 and Ile15 are highlighted in pink in the interfaces between constituent arcs that comprise the 20-meric NaD1–PA complex. **b** A close-up view of one of the isoleucine zippers between two arcs in the 20-meric complex. The seven pairs of isoleucines form a continuous hydrophobic interface between the two arcs as shown by the surface representation in pink

antitumor molecules. The ability of defensins to oligomerize is being increasingly recognized as a critical feature underlying these activities[18,19,21,24,30,31]. The human defensins HD6 and HBD-1 have both been shown to oligomerize and form nets to trap bacteria[30,31], however, the underlying molecular mechanism of action and any involvement of ligands remain to be clearly defined. In plants, the ability of certain defensins to attack and permeabilize target membranes has been shown to be dependent on the recognition of specific membrane phospholipids such as PIP$_2$ or PA, and subsequent formation of oligomeric defensin–phospholipid complexes[18,19,21]. Binding of PIP$_2$ by the plant defensins NaD1 and TPP3 results in defensin oligomerization, membrane permeabilization, and killing of tumor cells as well as fungi[18,20]. PC and PE are also membrane components of *C. albicans* and it is important to note that NaD1 does not bind either of these lipids, and does not permeablize PC containing liposomes[20]. HBD-3 has also been shown to bind PIP$_2$, resulting in the permeabilization of tumor cells, however the structural basis for this activity remains to be established[15]. These data suggest a conserved phospholipid-dependent mechanism of action in membrane permeabilization for defensins across kingdoms and species.

Although both NaD1–phospholipid complexes rely on cooperative binding of PIP$_2$ or PA, respectively, the mode of assembly differs substantially. Since PA lacks the inositol head group, residues His33 and Lys4 that are involved in PIP$_2$ binding[18] play no part in PA binding. Whereas NaD1–PA employed Arg39 as a key mediator of dimer–dimer interactions via PA, the same residue in the NaD1–PIP$_2$ complex is not used for any protein–protein or protein–PIP$_2$ interaction. Indeed, the use of Arg39 appears to be a key feature for the assembly of PA complexes for both NaD1 and NsD7, with Arg39 in the NsD7–PA complex playing a critical role for oligomerization[19]. Interestingly, we observed that the interactions between Arg39 (and Lys36) with PA are mediated via a set of three ordered water molecules. This raises the possibility that the NaD1–PA complex

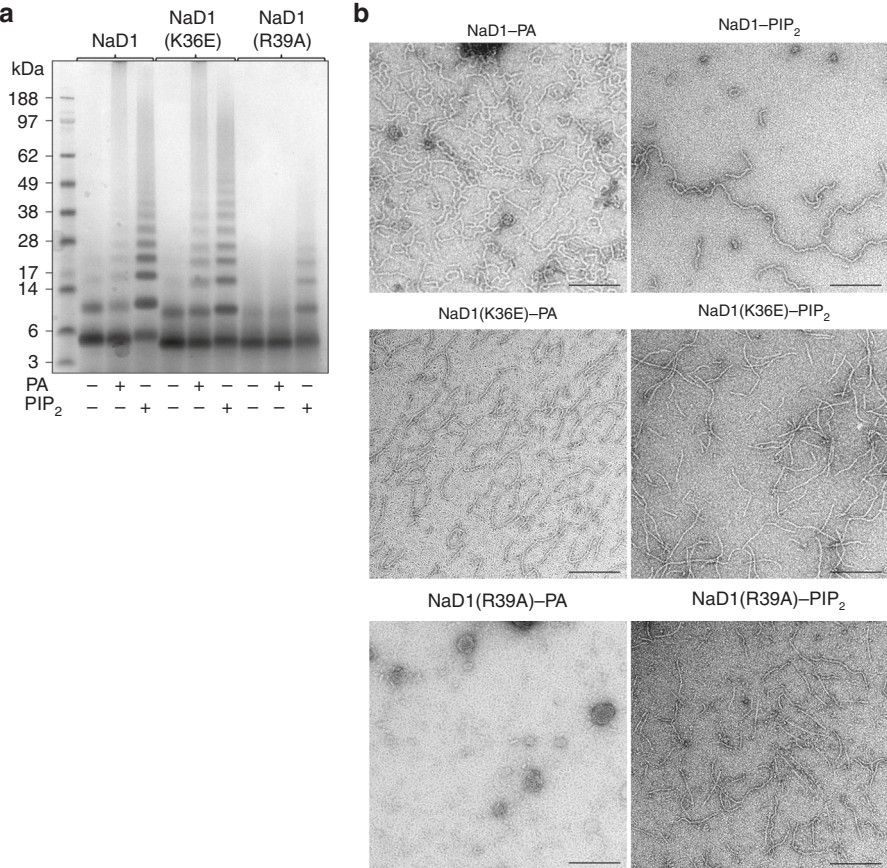

**Fig. 6** Site-directed mutagenesis of NaD1. **a** The ability of NaD1, NaD1(K36E), and NaD1(R39A) to form multimeric complexes in the presence of PA or PIP$_2$ as determined by protein–protein crosslinking with BS[3] and visualized by SDS-PAGE. **b** Transmission electron micrographs of NaD1, NaD1(K36E), and NaD1 (R39A) in the presence of PA or PIP$_2$. While PIP$_2$ binding to all three proteins resulted in fibril formation, PA-mediated fibril formation is abolished for NaD1 (R39A). Scale bars represent 200 nm

is weaker compared to the NaD1–PIP$_2$ complex, where key PIP$_2$ interactions responsible for cooperative binding and oligomerization are direct Arg–PIP$_2$ ionic interactions.

The relative ratio of PA to phosphatidylinositol content in the plasma membrane of *C. albicans* varies between strains and has been reported to range from 2:1 to 1:3 (PA:PI), comprising a total of >20% of the total phospholipid content of the membrane[33,34]. PIP$_2$ is expected to constitute ~5% of the total phosphatidylinositol amount in *C. albicans* as measured through intracellular myo [2-[3]H]inositol tagged phospholipids[35], which puts the relative PA:PIP$_2$ amounts between 40:1 and 6:1. Considering that these ratios vary between different *C. albicans* strains, it is not unexpected that we see some variation between the efficiency with which NaD1 as well as the NaD1(K36E) and NaD1(R39A) mutants kill *C. albicans*. Based on our chemical crosslinking data (Fig. 6a) we surmise that PIP$_2$ is a more potent inducer of NaD1 oligomerization compared to PA, since at equimolar concentrations of PIP$_2$ and PA we see more NaD1 captured in the crosslinker-induced laddering. Whilst this suggests that PIP$_2$ may be a more important driver for NaD1-mediated fungal killing, we nevertheless observed a significant reduction in *C. albicans* killing by the NaD1(R39A) mutant, which lost the ability to oligomerize in the presence of PA but still oligomerizes with PIP$_2$. Notably, two clinically relevant isolates of *C. albicans* were highly sensitive to NaD1 killing, but were highly resistant to the PA-specific NaD1(R39A) mutant.

NsD7 and NaD1 in complex with PIP$_2$ adopt near identical oligomeric topologies[21], however, PA-mediated oligomerization of NsD7 leads to a strikingly different topology compared to

NaD1–PA, where NsD7–PA adopts a double helical coiled structure by utilizing two distinct PA binding sites[19]. Furthermore, the NaD1–PIP$_2$ and NsD7–PA complexes only extend in one direction along the coil, whilst the NaD1–PA complex displays the ability to extend in two directions and is able to form a carpet-like structure. These features are likely due to the different use of the key isoleucine residues 15 and 17 in NaD1. Whereas in the NsD7–PA complex Ile15 and Ile17 are bound in a zipper-like structure at the core of the helical assembly, in the NaD1–PA complex both isoleucine residues are buried in the interface of neighboring dimers. This enables lateral association in a manner reminiscent of a Velcro strip, which permits joining of multiple dimers in a lateral fashion to form an extended carpet-like structure.

In a membrane environment it is envisaged that NaD1 dimers bind PA, and in turn recruit additional dimers through cooperative binding using Arg39 and Lys36 in the β2–β3 loop. The resulting arc-like oligomers then recruit additional arc-like complexes via the hydrophobic isoleucine 'Velcro' zipper. Since the MDC exhibits a slight curvature, we speculate that the formation of this complex is likely to induce curvature stress that would be expected to weaken the integrity of the target membrane (Fig. 8a). The combination of curvature stress and lipid sequestration is predicted to result in complete structural destabilization and subsequent permeabilization of the membrane. The key role of Arg39 is supported by functional assays that confirm that NaD1(R39A) is unable to oligomerize with PA, and has an attenuated ability to reduce growth and kill fungal cells. Although NaD1(R39A) was still an effective antifungal at higher

concentrations, this is to be expected as it is still able to bind to and oligomerize with $PIP_2$ which does not utilize Arg39[18]. Similar observations have been made for the legume defensin MtDef4, which has been shown to rely solely on PA binding for its antifungal activity[23]. Furthermore, our findings also suggest that NaD1 could form carpet-like structures with $PIP_2$. Alignment of

the NaD1–$PIP_2$ crystal structure with the central arc of the NaD1–PA crystal structure illustrates that although the specific interactions vary, the overall topology of the arcs remain the same (Fig. 8b). The NaD1–$PIP_2$ arcs have the same Ile13–Ile15 hydrophobic segment along its sides as in the NaD1–PA arcs and thus it is possible that a similar lateral extension of the

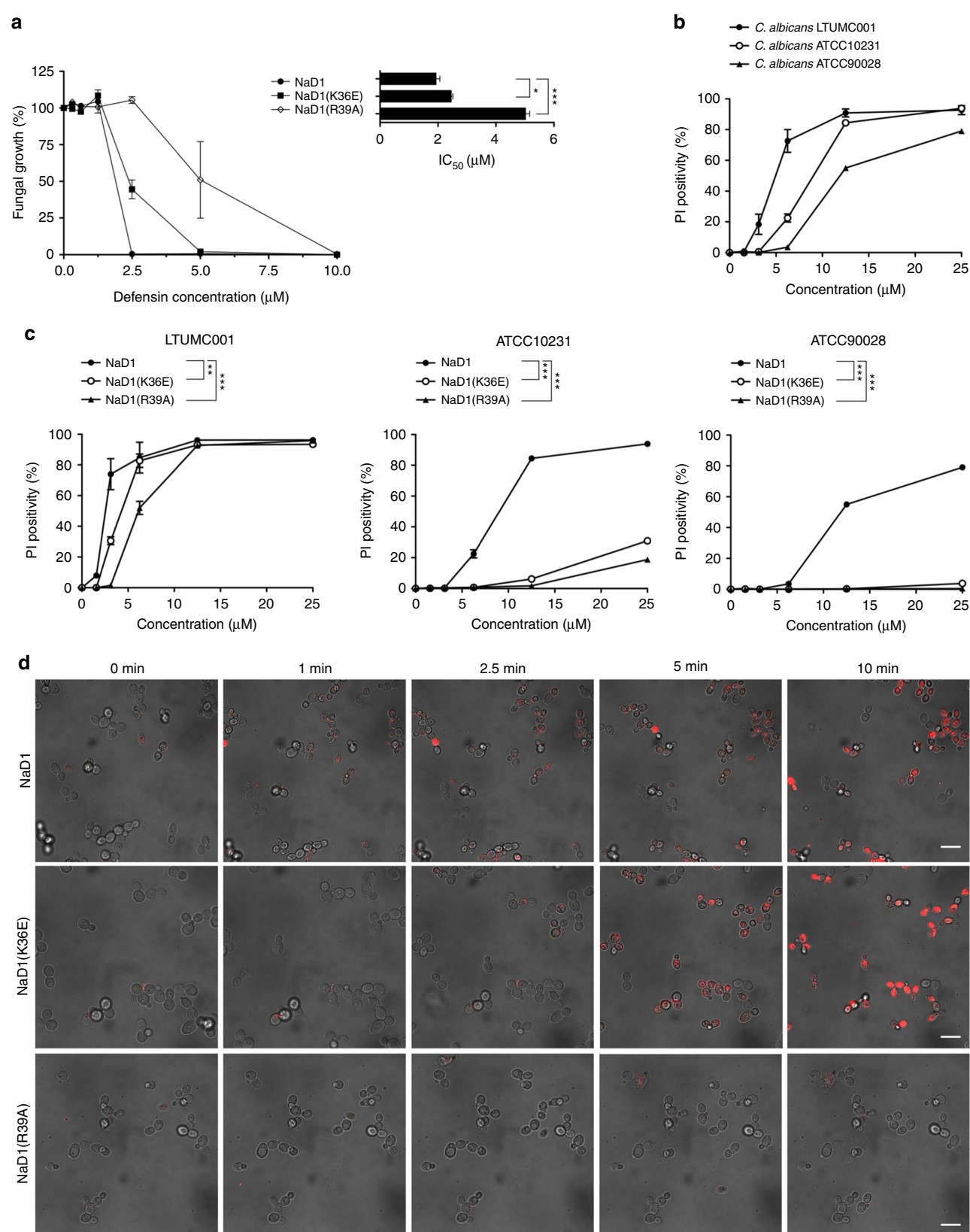

NaD1–PIP$_2$ complex via the recruitment of peripheral arcs would be achievable in vivo.

Our NaD1–PA complex structure provides a high-resolution view of a carpet-like mechanism of action for defensins. The carpet model was first proposed 25 years ago to rationalize the mechanism by which linear α-helical antimicrobial peptides permeabilize membranes[36], and was subsequently invoked as the leading mechanism of action for several of these types of CAPs[37]. Although the evidence for a carpet mechanism has been indirectly supported through a range of experiments[38], direct visualization at high resolution has proven to be elusive. Our findings provide clues to the likely configuration of NaD1 at the initial contact point with a phospholipid-bearing target membrane. It provides evidence that NaD1 is indeed able to adopt carpet-like oligomeric structures. In a broader context this strongly suggests that the phospholipid-binding capability of plant defensins greatly enhances their antimicrobial activity and relies on the formation of a diverse set of large oligomeric complexes with strikingly different architectures that ultimately mediate target membrane permeabilization.

## Methods

**Cloning and recombinant expression of NaD1 and its mutants**. DNA encoding the mature region of NaD1 (accession number AF509566) was amplified by PCR (Supplementary Table 1) and cloned into the pPIC9 expression vector (Invitrogen) directly in-frame with the α-mating factor secretion signal using the restriction enzymes XhoI and NotI. An alanine was added to the N-terminus of the NaD1 sequence to ensure efficient cleavage of the signal at the Kex2 cleavage site.

After transformation into E. coli TOP10 cells (Invitrogen), the pPIC9–NaD1 plasmid was isolated and linearized using SalI to allow integration at the his4 locus of the P. pastoris genome. Linearized DNA was transformed into electrocompetent P. pastoris strain GS115 cells (Invitrogen) and His$^+$ transformants were selected for by plating onto MD agar (13.4 g/L yeast nitrogen base [YNB], 400 µg/L biotin, 10 g/L dextrose and 15 g/L agar). A single His$^+$ colony was used to inoculate 200 mL of BMG (100 mM potassium phosphate, pH 6.0, 13.4 g/L YNB, 400 µg/L biotin, 1% [v/v] glycerol) and incubated with constant shaking at 30 °C until the OD$_{600}$ reached ~3.0. The cell mass was collected by centrifugation (1500g, 10 min) and resuspended into 1 L of BMM (100 mM potassium phosphate, pH 6.0, 13.4 g/L YNB, 400 µg/L biotin, 0.5% [v/v] methanol) to induce expression. Expression was continued for 72 h with constant shaking at 30 °C after which time the cell mass was removed by centrifugation (10,000g, 10 min) and the supernatant collected. One-twentieth volume of 1 M potassium phosphate buffer (pH 6.0) was added to the supernatant and the pH was adjusted to 6.0 with the addition of 10 M KOH. The supernatant was then applied to an SP Sepharose column (GE Healthcare Biosciences) pre-equilibrated with 100 mM potassium phosphate buffer (pH 6.0). Following extensive washing with 100 mM potassium phosphate buffer (pH 6.0), the bound proteins were eluted with 100 mM potassium phosphate buffer (pH 6.0) containing 0.5 M NaCl. The genes encoding the NaD1(K36E) and NaD1(R39A) mutants were commercially synthesized by GenScript (Piscataway, NJ, USA), subcloned into the pPIC9 vector and expressed and purified using the same protocol as used for wild-type NaD1. The purified proteins were concentrated using Amicon Ultra 3000 MWCO centrifugal filters (Millipore) and desalted into 10 mM MES pH 6.0. The protein concentration was determined using the BCA assay (Pierce). Protein purity and identity was confirmed by electrospray ionization–quadrupole–time of flight mass spectrometry analysis performed at the Comprehensive Proteomic Platform, La Trobe Institute for Molecular Science, La Trobe University.

**Crystallization and structure determination**. The NaD1–PA complexes were generated by mixing NaD1 with 08:0 PA (Avanti Polar Lipids) to a protein:lipid molar ratio of 1:2 (8 mg/mL:1.3 mg/mL). Crystals were grown over 5 days in sitting

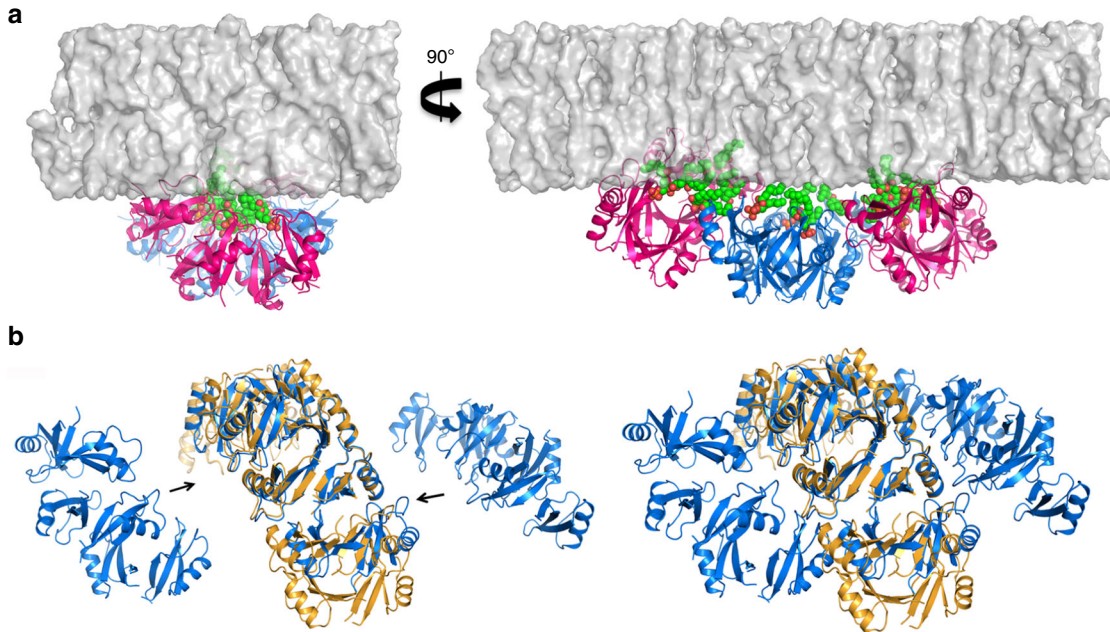

**Fig. 8** NaD1–PA oligomer assembly on a target membrane. **a** Cartoon representation of the membrane disruption complex (pink and blue), with PA molecules as spheres (red and green), as viewed on a flat model membrane (surface representation). The curvature of the membrane disruption complex is clearly seen. **b** Surface representation of the NaD1–PA complex (blue) aligned with the NaD1–PIP$_2$ complex (orange). In the left image, the two smaller arcs have been pulled apart from the central arc to highlight the similarities between the two structures, while in the right image both structures are shown unaltered

**Fig. 7** Fungal cell killing by NaD1 and mutants. **a** Fungal growth inhibition and estimated IC$_{50}$. C. albicans (LTUMC001) was incubated with titrations of NaD1 or mutant for 24 h, followed by OD$_{600}$ measurement to determine fungal growth. Data represent mean ± SEM of four independent experiments, performed in triplicate. *, $p < 0.05$; ***, $p < 0.01$, unpaired t-test. **b** Fungal cell permeabilization by NaD1 on various C. albicans isolates. **c** Fungal cell permeabilization by mutants in comparison to NaD1. C. albicans at $2.5 \times 10^6$ cells/mL in half-strength PDB was incubated with defensins at indicated concentrations. PI uptake was then determined by flow cytometry analysis. Data represent mean ± SEM of three independent experiments. ** $p < 0.01$; *** $p < 0.001$, Two-way ANOVA. **d** CLSM live imaging of NaD1 and mutants (5 µM) treated C. albicans (LTUMC001), immobilized onto an imaging chamber, in the presence of PI dye. Data are representative of three independent experiments. Scale bars represent 10 µm

drops at 20 °C by mixing 150 nL protein–lipid solution with 150 nL well solution containing 28% PEG 4000, 1 M sodium chloride, and 0.1 M trisodium-citrate-citric acid pH 5.9. Crystals were flash-cooled at 100 K in mother liquor supplemented with 20% ethylene glycol, and data were collected at the Australian Synchrotron (MX2) and processed using XDS[39]. The structure was solved by molecular replacement with PHASER[40] using the structure of NaD1 in its dimeric form[41] as a search model. The final model was built with Coot[42] and refined with Phenix[43] to a resolution of 2.5 Å with the final $R_{work}$ and $R_{free}$ values of 0.2031 and 0.2519, respectively. All data collection and refinement statistics are summarized in Table 1. All programs were accessed via the SBGrid suite[44]. Figures were prepared using PyMol. Movies were prepared using Chimera.

**Chemical crosslinking**. NaD1, NaD1(K36E), and NaD1(R39A) at 0.1 mM were incubated with or without 0.4 mM PA or $PIP_2$ at room temperature for 30 min. Protein complexes were crosslinked at room temperature for 30 min through primary amino groups by addition of 12.5 mM bis[sulfosuccinimidyl] suberate ($BS^3$) in a buffer containing 50 mM NaCl and 10 mM HEPES, pH 7.5. Reduced and denatured samples were subjected to SDS-PAGE and stained with InstantBlue (Expedeon).

**Transmission electron microscopy**. NaD1, NaD1(K36E), and NaD1(R39A) at 0.1 mM were incubated with 0.4 mM PA or $PIP_2$ at 4 °C for 24 h. Samples (10 μL) were applied to carbon-coated, 400-mesh copper grids, blotted with Whatman paper and twice negative stained with 7 μL 2% (w/v) uranyl acetate (Electron Microscopy Services), blotted as above and air-dried. Protein-only and lipid-only samples were prepared as controls. Micrographs were obtained using a JEM-2100 transmission electron microscope (JEOL) operated at 200 kV, equipped with a Gatan digital camera.

***Candida albicans* strains**. Clinical isolates of *C. albicans* ATCC10231 and ATCC90028 were obtained from the American Type Culture Collection (VA, United States) and LTUMC001 from the Department of Microbiology at La Trobe University (Melbourne, Australia).

**Fungal growth inhibition assays**. The ability of NaD1, NaD1(K36E) and NaD1 (R39A) to inhibit the growth of *C. albicans* was examined as described[45]. Briefly, *C. albicans* was grown overnight in yeast peptone dextrose (YPD) media (30 °C, 250 rpm). Spore number was determined using a hemocytometer, followed by dilution of the cell culture to $8 \times 10^3$ spores/mL in half-strength potato dextrose broth (Becton Dickinson). Diluted *C. albicans* spores (50 μL) were added to the wells of sterile 96-well flat-bottomed plates along with 50 μL of sterile-filtered defensin solution to final concentrations of 0–10 μM. After 24 h, cell growth was determined by measuring absorbance at 600 nm (9-well scan mode) in a SpectraMAX M5e plate reader (Molecular Devices).

**Propidium iodide uptake assay**. *C. albicans* strains were inoculated overnight in YPD media (30 °C, 160 rpm). Cells were pelleted, washed with $1 \times$ PBS buffer and counted using a haemocytometer prior to resuspending in half-strength potato dextrose broth (PDB; Becton Dickinson, NJ, USA). Cells ($5 \times 10^6$ cells/mL) were then treated with different concentrations of NaD1, NaD1(K36E) or NaD1(R39A) for 30 min at 30 °C, 160 rpm followed by propidium iodide (PI) staining (3 μg/mL; 5 min). The samples were diluted 5-fold with ice-cold PBS buffer prior to flow cytometry analysis.

**Determination of fungal colony forming units**. For survival assays, *C. albicans* overnight cultures were diluted to an $OD_{600}$ of 0.3 (~$5 \times 10^6$ cells/mL) in half-strength PDB and incubated for 5 h (30 °C, 250 rpm). Cells were then diluted to an $OD_{600}$ of 1.0 and treated with NaD1 or mutants (1.25, 6.25 and 25 μM) for 15 min (30 °C, 250 rpm). Survival was determined by counting CFU of 1/1000 and 1/10,000 dilutions after overnight growth on YPD plates at 30 °C.

**Confocal laser scanning microscopy**. Live imaging was performed on a Zeiss LSM-780 confocal microscope using a 63× oil immersion objective in a 37 °C incubator with 5% $CO_2$. *C. albicans* cells were immobilized onto 0.01% (w/v) poly-L-lysine-coated imaging chambers (LabTek, Hatfield, PA, USA) filled with half-strength PDB, containing 3 μg/mL PI. NaD1 or mutants were added directly to final concentration of 5 μM, via a capillary tube.

**Data availability**. Data supporting the findings of this manuscript are available from the corresponding authors upon reasonable request. The raw X-ray diffraction data were deposited at the SBGrid Data Bank[46] (http://data.sbgrid.org) as dataset entry 530 (doi:10.15785/SBGRID/ 530). The coordinates have been deposited in the Protein Data Bank (accession code 6B55).

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

## Acknowledgements
We thank the MX2 beamline staff at the Australian Synchrotron for help with X-ray data collection, and Janet Newman and Shane Seabrook at the Commonwealth Scientific and Industrial Research Organisation C3 Collaborative Crystallisation Centre for assistance with crystallization. We thank staff at the Comprehensive Proteomic Platform, La Trobe Institute for Molecular Science, La Trobe University, for assistance with mass spectrometry analyses. This work was supported by the Australian Research Council (Fellowship FT130101349 to M.K.).

## Author contributions
M.J.: Experimental design, acquisition of data; analysis and interpretation of data; and drafting and revising the article. F.T.L.: Acquisition of data; analysis and interpretation of data; and drafting and revising the article. T.K.P.: Acquisition of data; analysis and interpretation of data; and drafting and revising the article. C.H.: Acquisition of data; and analysis and interpretation of data. I.K.H.P.: Acquisition of data; and analysis and interpretation of data. M.R.B.: Acquisition of data; and analysis and interpretation of data. M.A.A.: Analysis and interpretation of data. M.D.H.: Conception and design; acquisition of data; analysis and interpretation of data; and drafting and revising the article. M.K.: Conception and design; acquisition of data; analysis and interpretation of data; and drafting and revising the article.

## Additional information

**Competing interests:** The authors declare no competing interests.

