## [Peer Review File · Nature Communications]

Reviewers' comments:

Reviewer #1 (Remarks to the Author):

This paper is well-written and it presents an interesting and significant structure. The paper could be stronger with additional data and revision to the text. However, I feel it is acceptable as a communication with only minor revision, if the editors agree.

The authors present the structure of NaD1, a plant defensin that they have previously crystallized with PIP2, in complex with phosphatidic acid. The asymmetric unit is a 20-mer of NaD1 with 14 phosphatidic acid molecules. The overall flat topology of the NaD1 20-mer suggests that NaD1 could assemble onto a membrane in a carpet-like fashion and the curvature of the 20-mer may help permeabilize the membrane, a key feature of many cationic antimicrobial peptides.

The structural analysis of the NaD1 20-mer is sound and well-explained. The overall architecture is flat, leading the authors to hypothesize that this is a membrane disruption complex for a carpet-like model of membrane disruption. I find this to be somewhat of a leap because there is little data presented that shows that this complex is required to permeabilize membranes. The mutagenesis data shows changes in the ability for NaD1 to form fibrils, but it is my understanding that these fibrils are distinct from the MDC presented in the paper. My suggestion would be to revise the ms. to make clear what is speculation versus strongly suggested by the data.

The authors argue that Arg39 is integral for oligomerization of NaD1. An R39A mutant was unable to form fibrils with PA but could still form fibrils with PIP2. However, *in vivo*, the NaD1 R39A mutant was still quite effective at killing *Candida albicans* (IC50 for WT is 2 μ M and IC50 for R39A is 4.5 μ M). The authors argue that this mutant "has a substantially attenuated ability to kill fungal cells." I don't find this to be the case. NaD1 can still kill *Candida albicans* effectively, perhaps because it can still interact with PIP2. This raises the question: does NaD1 prefer binding to PA over PIP2? What are the relative binding affinities of NaD1 to these two phospholipids, and consequently, what are the relative abundances of these lipids in the membranes of *Candida albicans*. Answering these questions could help explain why NaD1 R39A is unable to form fibrils with PA but can still kill *Candida albicans*.

The authors argue that hydrophobic arc-arc interactions are important for assembly of the MDC in Figure 5 and the associated text. I would like to see mutagenesis of Ile13 and Ile15 to prove that mutation of these residues effects fibril formation or fungal killing ability, although I do not consider it essential to publication.

Figure 7B would perhaps be better shown as a cartoon instead of a surface representation.

Reviewer #2 (Remarks to the Author):

The authors have investigated the interaction between the plant defensin NaD1 and PA using X-ray crystallography. The aim of the study was to get more insight in the defensin-mediated membrane permeabilization. To this end, the authors focused on the NaD1-PA interaction and on its impact on defensin-mediated fungal killing. This is an interesting study, there are however several shortcomings of the study as it stands now.

Although conclusions are drawn with regard to fungal killing and membrane permeabilization, both readouts have not been assessed. The authors should investigate membrane permeabilization *in vivo* in more detail by e.g. using specific dyes like propidium iodide and demonstrate that different doses of NaD1 result in compromised membranes whereas NaD1(R39A) doses do not. Moreover, not growth inhibition but killing should be assessed under these conditions.

Based on figure 6a, NaD1 (R39A) seems also to be characterized by reduced oligomerization in presence of PIP2. Hence the authors use TEM and assess fibril formation to discriminate between PA-mediated effects vs PIP-mediated effects of NaD1. Although the authors find a NaD1 R39A dose that only results in oligomers in presence of PIP2 and not in presence of PA, it still is highly speculative to draw firm conclusions toward PA-specific biological activity related to NaD1's oligomerization and growth inhibition. To what extent translates that specific dose (100 μ M) to differential inhibitory readout between NaD1 and NaD1(R39A)?

Fungal mutants characterized by altered levels of PA and/or PIP2 should be used to further complement this study and draw biologically meaningful conclusions.

Minor comment:

In general, the authors tend to generalize findings that only relate to specific defensins. It has been shown before that membrane permeabilization induced by certain defensins is a secondary effect and not the primary cause of growth inhibition. Generalizing statements in this regard should be omitted from the text.

Reviewer #3 (Remarks to the Author):

The findings in this ms highlights the ability of defensins to bind different types of phospholipids to form oligomeric fibrils with diverse topologies. This conceptual conclusion has been presented by the authors in previous papers (FEBS, PNAS, JBC). It has also been described that defensins assemble into nanonets (Chu et al., Science, 2012).

The major advancement of this paper, compared with the previously reported ones is the identification of a "carpet like" antimicrobial defensin-PA complex at 2.5 Å resolution. This is indeed a striking finding, also considering the high conservation between the different plant defensins.

The crystal structure determination is of good quality. However, there are some questions that can be raised.

1. Fig. 1 and 6. The authors base their analyses on generation of multimers using a concentration of 100 μ M defensin. This is followed by chemical crosslinking with BS. The buffer is 50 mM NaCl. One concern here is whether these conditions could induce artefacts in the system due to the high concentrations and use of crosslinkers. Have the authors seen the same oligomerization using native gels? EM could also be performed on non crosslinked material. The same question applies to the data reported in Fig. 6.

2. The conclusion that the R39A mutant is less active is based on tests with one *Candida* isolate. It is suggested that the authors provide additional data on other *Candida* isolates, preferably 6-8 isolates. The used *Candida* species is not defined in the ms. Was it an ATCC isolate? As the observation on the R39A mutation is a prerequisite for oligomerization and fungal killing it would also be valuable if the authors could add more data on the effects on *Candida*. Readouts could be i) live-dead assays, ii) EM analysis of membrane disintegration (any differences observed depending on the defensin used?)

3. PC, PE, PI, and PA etc are membrane components of *Candida*. Have the authors looked at effects of the defensin variants on liposomes composed of certain lipids (eg PC, PE or PA), and whether there is a correspondence in permeabilisation (see eg van der Weerden, JBC, 2010). This would further strengthen the story.

Response to reviewers' comments:

Reviewer #1:

Comment: The structural analysis of the NaD1 20-mer is sound and well-explained. The overall architecture is flat, leading the authors to hypothesize that this is a membrane disruption complex for a carpet-like model of membrane disruption. I find this to be somewhat of a leap because there is little data presented that shows that this complex is required to permeabilize membranes. The mutagenesis data shows changes in the ability for NaD1 to form fibrils, but it is my understanding that these fibrils are distinct from the MDC presented in the paper. My suggestion would be to revise the ms. to make clear what is speculation versus strongly suggested by the data.

Response: We have amended the manuscript throughout to make it clearer to the reader what is speculation versus what is supported by our data. For example, we amended the discussion on page 9: *"Since the MDC exhibits a slight curvature, we speculate that the formation of this complex is likely to induce curvature stress that would be expected to weaken the integrity of the target membrane (Fig. 8 a)."*

Also in the discussion on page 10: *"In a broader context this strongly suggests that the phospholipid-binding capability of plant defensins greatly enhances their antimicrobial activity and relies on the formation of a diverse set of large oligomeric complexes with strikingly different architectures that ultimately mediate target membrane permeabilization."*

We also have additional experimental data, which further strengthen our conclusions (see below). Furthermore, in support of our conclusions that a membrane disruption complex of NaD1 is important in cell permeabilization, we have included new additional data demonstrating the reduced ability of NaD1 mutants to permeabilize three different clinical isolates of *C. albicans* by measuring the uptake of the membrane impermeable dye propidium iodide by flow cytometry and live imaging by confocal laser scanning microscopy. These new data are presented in Figure 7 along with a description in the results on page 7 and discussion on page 8.

Comment: The authors argue that Arg39 is integral for oligomerization of NaD1. An R39A mutant was unable to form fibrils with PA but could still form fibrils with PIP2. However, in vivo, the NaD1 R39A mutant was still quite effective at killing *Candida albicans* (IC50 for WT is 2 uM and IC50 for R39A is 4.5 uM). The authors argue that this mutant "has a substantially attenuated ability to kill fungal cells." I don't find this to be the case. NaD1 can still kill *Candida albicans* effectively, perhaps because it can still interact with PIP2. This raises the question: does NaD1 prefer binding to PA over PIP2? What are the relative binding affinities of NaD1 to these two phospholipids, and consequently, what are the relative abundances of these lipids in the membranes of *Candida albicans*. Answering these questions could help explain why NaD1 R39A is unable to form fibrils with PA but can still kill *Candida albicans*.

Response: The reviewer has made an excellent point and we have endeavoured to address this issue with the following:

(i) We have included an additional paragraph in the manuscript where we discuss the relative abundance of PA and PIP₂ in *Candida albicans* membranes and implications in the oligomerisation and fungal killing by NaD1 (see page 8). Whilst we were unable to directly measure affinities of NaD1 to PA and PIP₂ due to the immediate oligomerization after phospholipid addition and cooperative nature of the binding, we examined the relative ability of PA and PIP₂ to induce NaD1 oligomerization. At similar concentrations, NaD1 treated with PIP₂ displayed substantially more laddering than after addition of a similar amount of PA, suggesting that PIP₂ is a more potent oligomer inducer and possibly has a higher affinity (although we cannot verify this latter point directly).

(ii). As highlighted above, we have also extended our analysis of the antimicrobial activity of NaD1 and R39A mutant against *Candida albicans* by including new data that examines direct fungal killing using PI uptake assays on the originally used strain as well as two other clinically relevant isolates, and live cell imaging in addition to the already presented growth inhibition assays. Importantly, these data show that two clinically relevant isolates of *C. albicans* were highly sensitive to NaD1 killing, but were highly resistant to the PA-specific NaD1 K39E mutant.

(iii). We have also amended the following sentence on page 9 in the discussion: "*The key role of Arg39 is supported by functional assays that confirm that NaD1(R39A) is unable to oligomerize with PA, and has an attenuated ability to reduce growth and to kill fungal cells.*"

Comment: Figure 7B would perhaps be better shown as a cartoon instead of a surface representation.

Response: We have amended Figure 7B and now show the structures as cartoon representation

Reviewer #2:

Comment: Although conclusions are drawn with regard to fungal killing and membrane permeabilization, both readouts have not been assessed. The authors should investigate membrane permeabilization in vivo in more detail by e.g. using specific dyes like propidium iodide and demonstrate that different doses of NaD1 result in compromised membranes whereas NaD1(R39A) doses do not. Moreover, not growth inhibition but killing should be assessed under these conditions.

Response: We agree and thank the reviewer for the helpful suggestion. We have now performed fungal killing assays using a FACS based method to evaluate killing directly via PI uptake to complement our data on growth inhibition, and further support these data with live CSLM microscopy. We have also extended these killing and membrane permeabilization assays to an additional two clinical isolates of *C. albicans*. These new data unequivocally show that clinically relevant isolates of *C. albicans* are highly sensitive to NaD1 killing, but resistant

to the PA-specific NaD1 K39E mutant (show little or no membrane permeabilizing activity). The new data is presented in Figure 7 with an accompanying description in the results on page 7 and discussion on page 8.

Results: “ We next examined whether the NaD1 induced growth inhibition of *C. albicans* is due to fungal killing by measuring uptake of the membrane impermeable nucleic acid dye propidium iodide in the presence of NaD1. We examined *C. albicans* LT1021 and an additional two clinical isolates ATCC10231 and ATCC90028, which all displayed significant PI uptake when treated with low μ M concentrations of wild type NaD1 (**Fig. 7 b**). To determine if the NaD1(R39E) and NaD1(K36E) mutants had impaired ability to kill *C. albicans*, we then examined their effect on PI uptake in the three test strains. Both mutants showed significantly reduced ability to permeabilize all three *C. albicans* isolates, with essentially no or low activity against ATCC10231 and ATC90028, although they retained some activity against LTU021 which was still significantly reduced when compared to WT NaD1 (**Fig. 7 c**). We then confirmed the PI uptake results by performing live confocal laser scanning microscopy, where we observed rapid PI uptake in *C. albicans* cells when treated with WT NaD1, whereas both NaD1 K36E and R39A showed substantially delayed and reduced PI uptake (**Fig. 7 d**). ”

Discussion: “ The relative ratio of PA to phosphatidylinositol content in the plasma membrane of *C. albicans* varies between strains and has been reported to range from 2:1 to 1:3 (PA:PI), comprising a total of >20% of the total phospholipid content of the membrane ^{33,34}. PIP₂ is expected to constitute ~5% of the total phosphatidylinositol amount in *C. albicans* as measured through intracellular myo[2-³H]inositol tagged phospholipids ³⁵, which puts the relative PA:PIP₂ amounts between 40:1 and 6:1. Considering that these ratios vary between different *C. albicans* strains, it is not unexpected that we see some variation between the efficiency with which NaD1 as well as the mutants NaD1K36E and R39E kill *C. albicans*. Based on our chemical cross-linking data (**Fig. 6 a**) we surmise that PIP₂ is a more potent inducer of NaD1 oligomerization compared to PA, since at equimolar concentrations of PIP₂ and PA we see more NaD1 captured in the cross-linker induced laddering. Whilst this suggests that PIP₂ may be a more important driver for NaD1 mediated fungal killing, we nevertheless observed a significant reduction in *C. albicans* killing by the NaD1 mutant K39E, which lost the ability to oligomerize in the presence of PA but still oligomerizes with PIP₂. Notably, two clinically relevant isolates of *C. albicans* were highly sensitive to NaD1 killing, but were highly resistant to the PA specific NaD1 K39E mutant. ”

Comment: Based on figure 6a, NaD1 (R39A) seems also to be characterized by reduced oligomerization in presence of PIP₂. Hence the authors use TEM and assess fibril formation to discriminate between PA-mediated effects vs PIP-mediated effects of NaD1. Although the authors find a NaD1 R39A dose that only results in oligomers in presence of PIP₂ and not in presence of PA, it still is highly speculative to draw firm conclusions toward PA-specific biological activity related to NaD1's oligomerization and growth inhibition. To what extent translates that specific dose (100 μ M) to differential inhibitory readout between NaD1 and NaD1(R39A)?

Response: For visualization of NaD1 oligomers using SDS-PAGE and TEM our lower limit for robust detection was 100 μ M. At lower concentrations both imaging using TEM and staining

of SDS-PAGE are difficult. However, we see differential inhibitory readout between NaD1 and NaD1(R39E) at much lower levels in growth inhibition assays as well as our additional killing assays (see Figure 7). Whilst there is a certain element of speculation until NaD1-PA oligomers are directly detected *in situ* on a fungal membrane, we believe our data (in particular the new fungal killing assay as described above) support the notion that there is significant PA-specific biological activity. This is evidenced clearly in our fungal killing assay using the *C. albicans* strains ACTT90028 and ATCC10231, where the NaD1R39E mutant that no longer binds PA and is unable to oligomerize (whilst retaining PIP₂ binding and forms oligomers with PIP₂) displayed a greatly reduced ability to kill.

Comment: Fungal mutants characterized by altered levels of PA and/or PIP₂ should be used to further complement this study and draw biologically meaningful conclusions.

Response: We agree with the reviewer that analysis of fungal mutants would help to support our conclusions, however we were unable to source such mutants with experimentally demonstrated altered PA and/or PIP₂ levels. We hope that our new additional data that more clearly demonstrates the important role of PA-binding by NaD1 in fungal cell killing by using additional clinically relevant isolates of *Candida albicans*, helps to address the reviewer's concerns.

Comment: In general, the authors tend to generalize findings that only relate to specific defensins. It has been shown before that membrane permeabilization induced by certain defensins is a secondary effect and not the primary cause of growth inhibition. Generalizing statements in this regard should be omitted from the text.

Response: We have amended statements of generalization to make it clear that the relevant findings relate to specific defensins and not all antimicrobial peptides. Specifically, we have amended the following sections:

Abstract: *"These findings identify a putative defensin–phospholipid membrane attack configuration that supports the longstanding proposed carpet mode of defensin peptide membrane disruption."*

We also amended the discussion on page 7: *"In plants, the ability of certain defensins to attack and permeabilize target membranes has been shown to be dependent on the recognition of specific membrane phospholipids such as PIP₂ or PA, and subsequent formation of large oligomeric defensin–phospholipid complexes."*

As well as on page 10: *"Our findings provide clues to the likely configuration of NaD1 at the initial contact point with a phospholipid-bearing target membrane."*

And finally on page 10: *"It provides evidence that NaD1 is indeed able to adopt carpet-like oligomeric structures."*

Reviewer #3:

Comment: Fig. 1 and 6. The authors base their analyses on generation of multimers using a concentration of 100 uM defensin. This is followed by chemical crosslinking with BS. The buffer is 50 mM NaCl. One concern here is whether these conditions could induce artefacts in the system due to the high concentrations and use of crosslinkers. Have the authors seen the same oligomerization using native gels? EM could also be performed on non crosslinked material. The same question applies to the data reported in Fig. 6.

Response: The reviewer raises an important point. Yes, we have seen oligomerization on non-reducing gels, and have added a new Supplementary Figure 1 in the supplementary material. We note that all negative stain EM was performed in the native state of the proteins without addition of cross-linkers. All oligomers observed by EM were solely due to the presence of lipid. We apologize if this was not clear in the manuscript, and have amended the relevant section in the results on page 5 as follows: *“In the presence of PA, NaD1 formed large oligomeric assemblies as shown by the distinct laddering after treatment with BS³ as well as on non-reducing SDS-PAGE (Supplementary Fig. 1). Importantly, native NaD1:PA complexes were visualized as long fibrils by TEM in the absence of any cross-linker (Fig. 1 b).”*

Comment: The conclusion that the R39A mutant is less active is based on tests with one Candida isolate. It is suggested that the authors provide additional data on other Candida isolates, preferably 6-8 isolates. The used Candida species is not defined in the ms. Was it an ATCC isolate? As the observation on the R39A mutation is a prerequisite for oligomerization and fungal killing it would also be valuable if the authors could add more data on the effects on Candida. Readouts could be i) live-dead assays, ii) EM analysis of membrane disintegration (any differences observed depending on the defensin used?)

Response: We thank the reviewer for identifying that omission, and the suggestion of including the analysis of additional *Candida albicans* isolates. We have expanded our study by including an additional two clinically relevant isolates sourced from ATCC that has strengthened our conclusions. All isolates, including the one used in the original experiments, have now been fully specified to allow unambiguous identification of the strain. Furthermore, we have added additional FACS based killing assays using PI uptake to support our initial data on growth inhibition. The new data on the additional two *C. albicans* isolates are presented in Figure 7, with accompanying new paragraphs in the results on page 7 and the discussion on page 8:

Results: *“We next examined whether the NaD1 induced growth inhibition of C. albicans is due to fungal killing by measuring uptake of the membrane impermeable nucleic acid dye propidium iodide in the presence of NaD1. We examined C. albicans LT1021 and an additional two clinical isolates ATCC10231 and ATCC90028, which all displayed significant PI uptake when treated with low μ M concentrations of wild type NaD1 (Fig. 7 b). To determine if the NaD1(R39E) and NaD1(K36E) mutants had impaired ability to kill C. albicans, we then examined their effect on PI uptake in the three test strains. Both mutants showed significantly reduced ability to permeabilize all three C. albicans isolates, with essentially no or low activity against ATCC10231 and ATC90028, although they retained some activity against LTU021 which was still significantly reduced when compared to WT NaD1 (Fig. 7 c). We then confirmed the PI uptake results by performing live confocal laser scanning microscopy, where we*

observed rapid PI uptake in C. albicans cells when treated with WT NaD1, whereas both NaD1 K36E and R39A showed substantially delayed and reduced PI uptake (Fig. 7 d)."

Discussion: "Whilst this suggests that PIP₂ may be a more important driver for NaD1 mediated fungal killing, we nevertheless observed a significant reduction in C. albicans killing by the NaD1 mutant K39E, which lost the ability to oligomerize in the presence of PA but still oligomerizes with PIP₂. Notably, two clinically relevant isolates of C. albicans were highly sensitive to NaD1 killing, but were highly resistant to the PA specific NaD1 K39E mutant."

Comment: PC, PE, PI, and PA etc are membrane components of Candida. Have the authors looked at effects of the defensin variants on liposomes composed of certain lipids (eg PC, PE or PA), and whether there is a correspondence in permeabilisation (see eg van der Weerden, JBC, 2010). This would further strengthen the story.

Response: We have previously shown that NaD1 does not bind PE or PC, and does not permeabilize membranes containing PC (see Baxter et al Mol Cell Biol 2015), whereas membranes containing PIP₂ are permeabilized. To cover this point, we have added the following statement to the discussion on page 7:

"PC and PE are also membrane component of C. albicans and it is important to note that NaD1 does not bind either of these lipids, and does not permeablize PC containing liposomes²⁰."

REVIEWERS' COMMENTS:

Reviewer #1 (Remarks to the Author):

The ms. is now acceptable, and the authors should be congratulated on a very nice piece of work!

Reviewer #2 (Remarks to the Author):

The authors have adapted the manuscript according to my comments. Only one comment remains: the authors should assess peptide-induced killing of *C. albicans* by cfu counting (and not rely on PI staining alone, which is a measure for general membrane permeabilization).

Reviewer #3 (Remarks to the Author):

The authors have made the appropriate amendments to the manuscript, and this referee is satisfied with the manuscript in its present revised form.

Note: native gels refers to the use of non-denaturing conditions (no SDS, see <https://www.ncbi.nlm.nih.gov/pmc/articles/PMC2898288/>). However, this is not a major concern here given the other supportive data, and the changes made in the present version are sufficient.

Response to reviewer comment:

Reviewer #2:

Comment: The authors have adapted the manuscript according to my comments. Only one comment remains: the authors should assess peptide-induced killing of *C. albicans* by cfu counting (and not rely on PI staining alone, which is a measure for general membrane permeabilization).

Response: We have now included assays using colony forming unit counts to examine NaD1 induced killing of *C. albicans*. The new data are featured in a new Supplementary Figure 7. We have also amended the results on page 7 based on these data: *“We next examined whether the NaD1 induced growth inhibition of C. albicans is due to fungal killing by measuring uptake of the membrane impermeable nucleic acid dye propidium iodide as well as colony forming assays, in the presence of NaD1. We examined C. albicans LTUMC001 and an additional two clinical isolates ATCC10231 and ATCC90028, which all displayed significant PI uptake when treated with low μ M concentrations of wild-type NaD1 (Fig. 7 b) and substantial reductions of colony forming units (Supplementary Fig. 7). To determine if the NaD1(R39E) and NaD1(K36E) mutants had impaired ability to kill C. albicans, we then examined their effect on PI uptake in the three test isolates. Both mutants showed significantly reduced ability to permeabilize all three C. albicans isolates, with essentially no or low activity against ATCC10231 and ATC90028, although they retained some activity against LTUMC001, which was still significantly reduced when compared to wild-type NaD1 (Fig. 7 c). Similar results were observed for the number of colony forming units (Supplementary Fig. 7). “*